# Different Roles between PEDOT:PSS as Counter Electrode and PEDOT:Carrageenan as Electrolyte in Dye-Sensitized Solar Cell Applications: A Systematic Literature Review

**DOI:** 10.3390/polym15122725

**Published:** 2023-06-18

**Authors:** Euis Siti Nurazizah, Annisa Aprilia, Risdiana Risdiana, Lusi Safriani

**Affiliations:** Department of Physics, Faculty of Mathematics and Natural Science, Universitas Padjadjaran, Jl. Raya Bandung-Sumedang KM 21, Sumedang 45363, Indonesia; euis12001@mail.unpad.ac.id (E.S.N.); a.aprilia@phys.unpad.ac.id (A.A.); risdiana@phys.unpad.ac.id (R.R.)

**Keywords:** counter electrode, DSSC, electrolyte, PEDOT:Carrageenan, PEDOT:PSS

## Abstract

Poly(3,4-ethylenedioxythiophene):polystyrene sulfonate (PEDOT:PSS) has been mostly used as a counter electrode to give a high performance of dye-sensitized solar cell (DSSC). Recently, PEDOT doped by carrageenan, namely PEDOT:Carrageenan, was introduced as a new material to be applied on DSSC as an electrolyte. PEDOT:Carrageenan has a similar synthesis process as PEDOT:PSS, owing to their similar ester sulphate (-SO_3_H) groups in both PSS and carrageenan. This review provides an overview of the different roles between PEDOT:PSS as a counter electrode and PEDOT:Carrageenan as an electrolyte for DSSC applications. The synthesis process and characteristics of PEDOT:PSS and PEDOT:Carrageenan were also described in this review. In conclusion, we found that the primary role of PEDOT:PSS as a counter electrode is to transfer electrons back to cell and accelerate redox reaction with its superior electrical conductivity and high electrocatalytic activity. PEDOT:Carrageenan as an electrolyte has not shown the main role for regenerating the dye sensitized at the oxidized state, probably due to its low ionic conductivity. Therefore, PEDOT:Carrageenan still obtained a low performance of DSSC. Additionally, the future perspective and challenges of using PEDOT:Carrageenan as both electrolyte and counter electrode are described in detail.

## 1. Introduction

In 1991, the dye-sensitized solar cell (DSSC) was presented by Grätzel and O’Regan for the first time [1,2] as promising third-generation solar cells, owing to the low cost of raw materials [3,4,5], the simple fabrication process [6,7] and relatively good energy conversion efficiency [8,9,10]. The DSSC is composed of a dye adsorbed with a semiconductor oxide as photoanode, an electrolyte and a counter electrode material [11]. The electron transfer reaction is commonly used in the DSSC working principle. Figure 1 shows the working scheme for DSSC. The energy of photon (hυ) is first absorbed by a dye that causes the electrons are excited from the highest occupied molecular orbital (HOMO) level to the lowest un-occupied molecular orbital (LUMO) level of dye. Then, the excited electrons are transferred to the external circuit through the conduction band of photoanode material. From the external circuit, the electrons enter again to the cell through a counter electrode material and then they move to the energy level of electrolyte for regenerating electrons in the dye through reduction-oxidation (redox) reaction. The current is formed by the moving electrons from dye and back again to dye successfully through all the process and it is repeated to observe the performance or efficiency of DSSC.

The series of chemical reactions that occur in DSSC are listed in Equations (1)–(5) [12]:(1)$$\mathrm{Anode}:\;D+h\upsilon \to D^{*}\;\mathrm{Absorption}$$
(2)$$\;D^{*}\to D^{+}+e^{-}\;\mathrm{Electron}\;\mathrm{injection}$$
(3)$$\;2D^{*}+3I^{-}\to 2D+I_{3}^{-}\;\mathrm{Regeneration}$$
(4)$$\;\mathrm{Cathode}:\;I_{3}^{-}+2e^{-}\to 3I^{-}\;\mathrm{Reduction}/\mathrm{Recaption}$$
(5)$$\mathrm{Cell}:\;e^{-}+h\upsilon \to e^{-}\;$$

The development research of DSSC has been focusing on increasing the efficiency with the great performance of DSSC. One of the materials that have been used in sandwich structure DSSC both as a counter electrode and an electrolyte is a conductive polymer. Conductive polymers are a group of organic polymers with π-conjugated structure that possess the electrical (high intrinsic conductivity) and optical properties while maintaining the mechanical properties [13]. The unique properties of conductive polymer, i.e., good optical properties, high conductivity, lightweight, low cost, flexibility and excellent processability, in industrial manufacturing are utilized in DSSC. One of the conductive polymers that have been used in DSSC as a counter electrode and an electrolyte is poly(3,4-ethylenedoxythiophene) (PEDOT).

PEDOT is one of the derivates of polythiophene that has been drawing attention due to its excellent conductivity, good transparency, good stability, low oxidation potential and low band gap [1,11]. It is soluble in many solvents due to the existence of polystyrene sulfonate (PSS) as a dopant. The PSS plays a crucial role in the stability of PEDOT:PSS dispersion, owing to its negative charge as the counter ion while doping to the PEDOT. PSS has ester sulphate (-SO_3_H) functional group as the hydrophilic part that assists the dispersion in PEDOT:PSS [14]. PEDOT:PSS is used mostly as a counter electrode with the efficiency from 2–7%, reportedly [3,4,8].

On the other hand, a similar thing has been found that in the presence of carrageenan, PEDOT is also dispersible in an aqueous solution. Carrageenan, with an ester sulphate (-SO_3_H) functional group which is the identical functional group as in PSS, is an anionic heteropolysaccharide extracted from seaweed [14]. Therefore, carrageenan could be another dopant for PEDOT to form PEDOT:Carrageenan. In recent research, PEDOT:Carrageenan is used as an electrolyte in DSSC with low efficiency.

The study of PEDOT and PEDOT:PSS related to their applications has been reviewed in several papers. Most of these review papers discuss the characteristics of PEDOT and PEDOT:PSS as a counter electrode of DSSC. L. Fagiolari et al. have offered an overview of PEDOT-based materials and proposed the use of them in replacing Pt as counter electrodes in DSSCs [15]. W. Wei et al. addressed the progress of PEDOT-based counter electrodes for DSSC with a specific discussion on the characteristics of PEDOT, which was composited with other materials, such as carbon, metal, metal oxide, etc. [16]. However, there are a few research articles that have been published for PEDOT:Carrageenan as an electrolyte for DSSC. Therefore, there is no comprehensive paper comparing PEDOT:PSS and PEDOT:Carrageenan in one paper related to their applications in DSSC. In this review article, we studied the different roles between PEDOT:PSS as a counter electrode and PEDOT:Carrageenan as an electrolyte in DSSC applications, respectively. This paper aims to compare the role of PEDOT:PSS as counter electrode and PEDOT:Carrageenan as electrolyte for DSSC applications.

## 2. Methods

This review gives the overall description of the use of PEDOT:PSS and PEDOT:Carrageenan for DSSC application, respectively. The literature search employed the following databases at Springer, IOPscience and Google Scholar with the keyword combination: (“PEDOT:PSS” AND “counter electrode”) OR (“PEDOT:Carrageenan” OR “PEDOT:Carr” AND “electrolyte”) OR (“polymer conductive” OR “polymer electrolyte”) AND (“DSSC” OR “Dye-Sensitized Solar Cell”). We restricted the search to articles published in international journals that were written in English and full-text available (open-access articles). The total documents were 1087 papers as shown in Table 1 in detail.

The overall identified documents, according to the years of publication, were limited from 2012 to 2022, with 612 files from Springer, 111 files from IOPscience and 222 files from Google Scholar. In order to exclude all documents other than research and conference papers, the type of paper screening was used, and 332 articles were received. Studies on the performance of DSSC with PEDOT:PSS as a counter electrode and PEDOT:Carrageenan as an electrolyte were included as inclusion criteria. The exclusion criteria include articles that just describe PEDOT:PSS or carrageenan in other applications. After screening the title and abstract, there were 47 articles that meet the inclusion criteria and 285 excluded articles were removed. Five of the 47 articles have been eliminated as a result of duplication, and 30 articles were removed because the data was incomplete for analysis. As result, there are 12 full-text articles to be analyzed in this review paper consisting of nine articles for PEDOT:PSS as counter electrode and three articles for PEDOT:Carrageenan as electrolyte for DSSC. Figure 2 shows the PRISMA guideline for selection procedures.

## 3. Synthesis and Characteristics of PEDOT:PSS and PEDOT:Carrageenan

### 3.1. Synthesis and Characteristics of PEDOT:PSS

In 1988, PEDOT with cation of EDOT monomer ${\left(C_{2}H_{4}O_{2}C_{4}H_{2}S\right)}^{+}$ was first synthesized and commercialized by scientists in a German research laboratory [13]. PEDOT has succeeded in becoming one of the electronically conducting polymers (ECPs) that have immense potential applications. However, PEDOT was difficult to be synthesized by the standard chemo/electro-polymerization due to its insolubility and infusibility in many solvents. Therefore, PEDOT needs some materials as a dopant, which is a solvent-dispersible material, while maintaining the properties of PEDOT.

PSS with a chemical formula of ${C_{8}H_{7}{SO}_{3}}^{-}$, is a water-dispersible polyelectrolyte and a polymer surfactant that is able to be used as a dopant for PEDOT due to its -SO_3_H functional group as the hydrophilic part. PEDOT and PSS formed a complex material as PEDOT:PSS, which is a poly–ion complex by electrostatic interaction between PEDOT cation and PSS anion. It consists of both positive charges conjugated PEDOT and negative charge saturated PSS. The chemical structure of PEDOT, PSS and PEDOT:PSS is shown in Figure 3, and the structure identification was completed with the FTIR analysis in Table 2. In PEDOT:PSS, PSS has two jobs: (i) it works as a counter ion for doped-PEDOT stability, and (ii) it supplies a matrix to form an aqueous dispersion [17].

PEDOT:PSS was also first synthesized by scientists in German research laboratories. PEDOT:PSS is the most successful commercially available in the form of an aqueous dispersion with high water-dispersibility, excellent miscibility, good transparency, high electrical conductivity, excellent flexibility and satisfactory stretch-ability. PEDOT:PSS is typically synthesized via oxidative polymerization in two simple ways: in-situ and post-polymerization [19,20]. The synthesis process is shown in Figure 4. In-situ polymerization has several steps to do as follows: First, monomer powder EDOT was added to an aqueous of PSS solution. Then, the mixtures were stirred vigorously in a water bath at room temperature under nitrogen. The oxidant agents (e.g., sodium persulfate (N_2_S_2_O_8_) and iron trichloride (FeCl_3_) or iron tri-sulfate (Fe_2_(SO_4_)_3_)) were immediately added to the mixture solution to produce a complex and stirred again at room temperature for 24 h. The precipitate was collected from the complex solution by centrifugation at a certain rpm and then rinsed with acetone and methanol at a certain ratio. The final result can be dispersed in water to be aqueous in a dark-blue solution or be dried in an oven at 60 °C for 24 h to get a black-powder. Post-polymerization includes the steps as follows: first, monomer powder EDOT was dispersed in water. The oxidant agents (e.g., N_2_S_2_O_8_ and FeCl_3_ or Fe_2_(SO_4_)_3_) were immediately added to EDOT solution and stirred at room temperature for 24 h. Then, the mixture was purified by mixing water and ethanol at a certain ratio. The PEDOT powder as a result was added to the PSS solution. The PEDOT:PSS solution was stirred at room temperature for 24 h. The final result was the PEDOT:PSS aqueous in a dark-blue solution.

The properties of PEDOT:PSS are crucial things to consider in the synthesis process, especially for electrical conductivity. There are some factors that affect the electrical conductivity of PEDOT:PSS as shown in Figure 5. In the synthesis process, some parameters should be controlled, such as pH solution, temperature, humidity and polar solvent as additives or blending components. PSS has hygroscopic and corrosive properties due to its strong acid (pH < 2), which lowers the lifetime and performance of the application devices. The neutralizing of PSS can be attained by various alkaline, but this could change the structural and electrical conductivity resulting in PEDOT:PSS [21].

Several studies showed that the higher the pH of the solution resulting from the synthesis process, for example, due to adding alkaline materials (e.g., sodium hydroxide (NaOH)), the more the electrical conductivity decreased and vice versa. If the resulting solution was in a lower pH, by adding acid materials (e.g., hydrochloride acid (HCl)) during the synthesis process, the electrical conductivity increased [13,21]. Mochizuki et al. have investigated the effect of adding NaOH concentration from 0.8 to 1.2 M with increasing pH of 2.5 to 11.7 to cause decreasing electrical conductivity of PEDOT:PSS from 10^−2^ to 10^−4^ S·cm^−1^ [21]. This is due to the removal of the insulting PSS from the surface of the colloidal particles and to the crystallization of the PEDOT molecule, which improves both intraparticle and interparticle transfer to charge carrier. The high electrical conductivity affected the higher carrier mobility and the structure of PEDOT:PSS. The lower pH and also adding polar solvents partially change PEDOT molecules with an amorphous form to a crystalline state by, i.e., ethylene glycol (EG), dimethyl sulfoxide (DMSO), dimethyl formamide (DMF), etc. that were added during the synthesis process. PEDOT in the crystalline state has an orthorhombic phase that represents π−π stacking of the PEDOT molecules with XRD peak at 2θ = 26° (020) [21,22,23].

Besides pH and polar solvent, the humidity and temperature affected the electrical conductivity of PEDOT:PSS in thin layer form. The PEDOT:PSS can be applied in many application devices if it was made in thin layer form. Several methods, such as spin-coating, screen-printing, electrospinning, etc., are used as a technique for thin layer deposition. The electrical conductivity of PEDOT:PSS layer was changed in water. At first, the electrical conductivity initially increased significantly, but then it gradually decreased when its layer was slowly damaged, depending on the longer immersed time of PEDOT:PSS layer in water. Otherwise, when the temperature went down, the electrical conductivity of PEDOT:PSS layer started to decrease [13,17].

### 3.2. Synthesis and Characteristics of PEDOT:Carrageenan

Carrageenan comes from various forms of red algae (*Rhodophyta*) and is used as a general name for a poly-saccharides (sulphated galactans) family. Chemically, carrageenan is a linear polymer, which is arranged of alternating *disaccharide* repeating units of 3-linked β-D-*galactopyranose* (G-units) and 4-linked α-D-*galactopyranose* (D-Units) or 4-linked 3,6-*anhydrogalactose* (DA-Units) [24,25]. In 1862, the British pharmacist Stanford found the first called *phycocolloid carrageenin* by extracting it from Irish moss (*Chandrus crispus*) [25]. The name was later changed to *carrageenan* so as to fulfill with the “-an” suffix the names of polysaccharides that were accepted in the industry dates from the 1940s [26].

Carrageenan has three commercially important types according to the position and number of sulphate groups: *kappa (κ)-*, *iota (ι)-* and *lambda (λ)-carrageenan*. The *κ-carrageenan* with the chemical formula of ${{C_{12}H_{17}O_{9}SO}_{3}}^{-}$ is produced from the seaweed *Kappaphycus alvarezii*, known as *Euchema cottonii* (or simply *cottonii*). The *ι-carrageenan* with the chemical formula of ${{{C_{12}H_{16}O_{9}(SO}_{3}}^{-})}_{2}$ is largely produced from *Euchema denticulum*, known as *Euchemaspinosum* (or simply *spinosum* species). The *λ-carrageenan* with the chemical formula of ${{{C_{12}H_{17}O_{10}(SO}_{3}}^{-})}_{3}$ is extracted from the species of *Gigantana* and *Chondrus ginera*. There are also several other *carrageenan* repeating units, e.g., *mu (µ)-*, *nu (υ)-*, *xi(ξ)-*, *theta (θ)- and beta (β)-carrageenans* [26,27,28]. Figure 6 shows all types of *carrageenan.* Based on IUPAC and the letter codes of carrageenans, which are shown in Table 3, their corresponding names of *κ-*, *ι-* and *λ-carrageenan* are *carrageenase 2,40-disulphate* (G4S-DA2S), *carrageenase 40-sulphate* (G4S-DA) and *carrageenan 2,6,20-trisulphate* (G2S-D2S,6S), respectively [26,27].

Carrageenan is one of the natural polymers, with ester sulphate (-SO_3_H) groups, that has the capability to produce thermo-reversible gels or high viscous solution. In a number of foods and pharmaceutical and cosmetic products, it is commonly used as gellifier, stabilizer and emulsifying agent [29]. Carrageenan has the similarity of ester sulphate (-SO_3_H) groups to PSS. PSS has been used as a dopant in complex material PEDOT:PSS; then, carrageenan also has the potential to be used as dopant to form a complex material PEDOT:Carrageenan. Figure 7 shows the complete chemical structure of PEDOT:Carrageenan, and the structure identification was completed with FTIR analysis in Table 4.

PEDOT:Carrageenan was synthesized via oxidative polymerization [14,30] as synthesis of PEDOT:PSS. The oxidative polymerization was done in two simple ways: in-situ and post-polymerization. The procedures of both two ways as similar as PEDOT:PSS, but the carrageenan powder was first diluted in water under stirring at 70 °C. Then, monomer EDOT, oxidant agents and surfactant were mixed together until the PEDOT:Carrageenan solution was obtained in homogenous solution. The process synthesis of PEDOT:Carrageenan is shown in Figure 8.

In summary, PEDOT:PSS and PEDOT:Carrageenan has similar synthesis process because both PSS and Carrageenan has ester sulphate (-SO_3_H) functional group that cause them as a dopant for PEDOT. However, PSS and Carrageenan have different polymer structure that affects the molecular weight of both PEDOT:PSS and PEDOT:Carrageenan. By analyzing the chemical structure of their monomer as shown in Figure 3 and Figure 7, the molecular weight of both PEDOT:PSS and PEDOT:Carrageenan could be calculated by adding up the molecular weight of their monomer. It depends on their chain numbers (n). PEDOT:PSS has EDOT monomer and styrene sulfonate (SS) monomer with a molecular weight of 142 and 183 g/mol, respectively. Three types of PEDOT:Carrageenan, namely PEDOT:κ-Carrageenan, PEDOT:ι-Carrageenan and PEDOT:λ-Carrageenan, consist of EDOT monomer and carrageenan monomer (κ-, ι- and λ-carrageenan). Molecular weight of κ-, ι- and λ-carrageenan monomer is 385, 464 and 561 g/mol, respectively.

Table 5 shows the difference in the electrical and optical properties of PEDOT:PSS and PEDOT:Carrageenan. PEDOT:PSS has excellent conductivity of >4000 S·cm^−1^ with a low sheet resistance of <100 Ω·sq^−1^ and high transparency of 80–95% [13]. With high conductivity, PEDOT:PSS has ionic and electronic mobility of 2.2 × 10^−3^ and 1.3 cm^2^v^−1^s^−1^, respectively [31], with carrier density of 4 × 10^20^ cm^−3^ at approximately +0.5 Volt [32]. However, PEDOT:Carrageenan has conductivity of 16.23 S·cm^−1^, and it was measured for PEDOT:κ-Carrageenan [33]. The value of sheet resistance, transparency, ionic and electron mobility and carrier density of PEDOT:Carrageenan has not been available since the research of PEDOT:Carrageenan is still very limited, especially in analyzing the electrical and optical properties that are the basis for the application of a material in electronic devices.

## 4. Results and Discussions

### 4.1. Conductive Polymer for DSSC

DSSC has a traditional structure that consists of a transparent photoanode with a dye-sensitized mesoporous thin layer, $I^{-}$/$I_{3}^{-}$ redox electrolyte and a counter electrode with a catalytic layer. Recently, there are many papers that researched how to increase the performance of DSSC, starting from the material selection for photoanode materials like some metal-oxide semiconductors, e.g., titanium dioxide (TiO_2_), zinc oxide (ZnO), zirconium oxide (ZrO_2_), etc., even up in composite form between two or more metal-oxide semiconductors such as a composite of TiO_2_:ZnO, TiO_2_:ZrO_2_, etc. Also, many researchers are starting to use natural dyes as dye-sensitized like extracted from leaves, vegetables, etc., which can be absorbed well by photoanode materials. As well as in the selected materials for electrolyte and counter electrode, where these two materials work together in electron transfer from the external circuit to the internal circuit for obtaining the current.

Conductive polymers have been mostly used as a counter electrode and an electrolyte system in DSSC. These two components have a crucial role to create charge transport electrons. The counter electrode is defined as the electrode at which electrons enter the cell and reduction occurs. Meanwhile, the electrolyte is sandwiched between the photoanode and the counter electrode to stabilize the operation of the DSSC; as for the regeneration of the dye and itself, it must carry the charge between the photoanode and the counter electrode.

In DSSC, the counter electrode has been responsible for three roles: (i) as a catalyst to accelerate the reduction reaction of redox pairs in the electrolyte, (ii) as a positive electrode to gather the electrons from the external circuit and transport them into the cell and (iii) as mirror or reflector to reflect the unabsorbed light from the cell back into the active area; hence, the absorption rate of sunlight into the DSSC is increased [34,35]. In order to reduce the redox couple, an ideal counter electrode material is expected to have a large surface area and a high porosity and a good catalytic activity within the electrolyte that has matched energy levels for redox couple potential, high electrical conductivity, good reflectively, low-cost, easy synthesis process, optimum thickness and strong adhesivity on TCO substrate, chemically, mechanically and electrochemically stable. Several papers have included a review about various types of conductive polymers as counter electrodes for DSSC. They said that the potency of conductive polymers to be a counter electrode material for DSSC had given the promise to replace Pt-based counter electrodes with high efficiency [36,37]. PEDOT:PSS is one of the counter electrode polymers that has been used in DSSC with a high performance of 8.49% [38].

Electrolytes have roles in DSSC as an electron transfer for the purpose of regenerating the dye sensitized from the oxidized state. The electrolyte must have long-term stability and high ionic conductivity. In general, electrolytes for DSSC are formed in liquid, quasi-solid and solid. Liquid electrolytes have been mostly used for DSSC due to easy fabrication. The utilization of a liquid electrolyte is capable of producing relevant technological problems, for example, limited long-term stability, trouble in tough and airtight sealing, evaporation and seepage of electrolyte in the event of the breaking of the glass substrates, platinized cathode corrosion, shift of the adsorbed dye, water and oxygen permeability [39]. To address this problem, the polymer could be used as an electrolyte due to its properties, such as high ionic conductivity, transparency, thin-film forming ability, flexibility and easy processability. Some papers have reviewed the polymer electrolytes for DSSC with high ionic conductivity of 4.75 × 10^−2^ S/cm [40,41]. Polymer electrolytes is classified into three different types based on their physical state and composition: (i) gel polymer electrolytes, (ii) solid polymer electrolytes and (iii) composite polymer electrolytes [12].

The conductive polymer as a counter electrode and electrolyte becomes interesting research since the properties of the polymer are matched to the criteria of ideal material in both the counter electrode and electrolyte. One of the conductive polymers that have been mostly used as a counter electrode and electrolyte is poly(3,4-ethylenedioxythiophene) (PEDOT) and their composites like PEDOT:PSS and PEDOT:Carrageenan.

### 4.2. PEDOT:PSS as Counter Electrode

The recent development research of PEDOT:PSS as a counter electrode for DSSC application is summarized in Table 6. There are factors that have affected the quality of PEDOT:PSS-based counter electrode in DSSC performance, either intrinsically or extrinsically. Intrinsically, the characteristics results should obtain the match properties of PEDOT:PSS as a counter electrode, such as electrical conductivity value and electrocatalytic activity capability. It was related to the chemical synthesis process that has done in each experiment. Extrinsically, the quality of the thin film can be seen from thickness, morphology and roughness, which was related to the treatment that was given during the thin film fabrication process.

The thin film of PEDOT:PSS with a different number of layers and temperature annealing can be affected to the performance DSSC. PEDOT:PSS thin film with three layers of coating at 120 °C resulted in the highest efficiency of 1.77% [42]. Increasing the number of layers and the annealing temperature of PEDOT:PSS film has enhanced the electrical conductivity value. However, the higher annealing temperature produced a thinner layer of PEDOT:PSS film and caused the decreasing PEDOT:PSS roughness. It hinders the electrocatalytic activity of PEDOT:PSS that causes low electron mobility. Therefore, increasing the number of layers and temperature of PEDOT:PSS film can cause low short-circuit current density (*J_sc_*) that the DSSC efficiency decreases.

PEDOT:PSS-based DSSC still has low efficiency with low *J_sc_* and fill factor (*FF*) due to the high series resistance of the cell that was caused by the poor contact between PEDOT:PSS and substrate. Polyethylene glycol (PEG), a water-soluble polymer with excellent film-forming, emulsifying and adhesive properties, can assist the binding strength between PEDOT:PSS and substrate. PEG also serves as a binder to improve the mechanical properties of the counter electrode. The highest efficiency of 4.39% was obtained by mixing the PEDOT:PSS and 5 wt% of PEG in 0.2 wt% of acetylene black [43]. It was proven that the best binding strength between PEDOT:PSS and substrate caused a low sheet resistance value that indicated high electrical conductivity and improved the *J_sc_* and *FF* values. These results enhanced the performance of DSSC using PEDOT:PSS as a counter electrode.

The stability of PEDOT:PSS during the preparation process is most important to keeping the good properties and quality of the counter electrode. PEDOT:PSS can be easily combined with many other materials and compounds, such as metal-oxide semiconductors, metal materials, sulfide compounds, etc. TiO_2_ is one of the metal-oxide semiconductors which can be mixed with PEDOT:PSS as a counter electrode. DSSC, with a mixture of PEDOT:PSS and TiO_2_ nanoparticles, has the highest efficiency of 8.49% [38]. For the improvement of film redox reaction and DSSC performance, TiO_2_ nanoparticles are available to increase polymer surface. Other materials, e.g., nickel (Ni), nickel sulfate (NiSO_4_) and nickel sulfide (NiS), can also be combined with PEDOT:PSS as a counter electrode. The composite of PEDOT:PSS with those materials, as well as the composite PEDOT:PSS with TiO_2_ nanoparticles, can enhance the electrical conductivity and electrocatalytic activity by increasing the large surface area and decreasing the sheet resistance. The highest efficiency of 2.25% and 3.05% could be obtained from the composite of Ni-PEDOT:PSS and NiSO_4_-PEDOT:PSS, respectively [44]. Meanwhile, the composite of NiS-PEDOT:PSS obtained the highest efficiency of 8.18% [45].

The combination of PEDOT:PSS and carbon materials can also be used as counter electrodes to enhance the performance of DSSC. Carbon materials with polymers enhanced mechanical and thermal stability, electrical conductivity and electrocatalytic activity due to increased surface area. PEDOT:PSS/Carbon composite was formed by mixing the PEDOT:PSS solution and a small amount of graphite powder. The composite film was coated on FTO using a scratch method at vacuum at 80 °C with a thickness of 3–4 μm. The highest efficiency of 7.60% was obtained due to the high electrical conductivity and high electrocatalytic activity in the $I^{-}/I_{3}^{-}$ redox reduction [3]. In the same way, PEDOT:PSS/Graphene composite obtained the highest efficiency of 4.66% [46]. Other carbon materials, such as single-wall carbon nano-horn (SWCNH) and single-wall carbon nanotube (SWCNT), can be composited with PEDOT:PSS. PEDOT:PSS/SWCNH bilayer films obtained the highest efficiency of 5.10%. The proposed bilayer is more valuable since it has a couple of merits of SWCNH like high specific surface area, excellent electrocatalytic ability and merits of PEDOT:PSS like high conductivity [47]. The performance of DSSC increased due to highly textured surface, large surface area, increased surface roughness, evidencing more numbers of catalytic active sites for reduction of $I_{3}^{-}$ to $I^{-}$ ions. Increasing surface area in case of bilayer counter electrode could be attributed to the compositional/stacked effect of PEDOT:PSS and SWCNH. Meanwhile, the combination of PEDOT:PSS and SWCNT with addition of molybdenum disulfide (MoS_2_) obtained the highest efficiency of 8.14%. MoS_2_ is transition metal sulfide, as a typical lamellar compound that has the similar structure to graphene. MoS_2_ can be prospective used as a lubricant hydrogen evolution reaction, electrode materials for lithium battery and DSSC [48]. The DSSC performance increased due to the high catalytic activity of counter electrode. The enhanced electrocatalytic activity corresponds to its synergetic catalytic effect, such as distinctive structure with large surface area of PEDOT:PSS, the improvement conductivity of SWCNT and excellent catalytic activity of MoS_2_, which could make electron transmit across the counter electrode/FTO interface easily [48].

### 4.3. PEDOT:Carrageenan as Electrolyte

Research related to PEDOT:Carrageenan for DSSC was found in some papers as presented in Table 7. In general, characteristics of carrageenan are similar to cellulose, which is always used as electrolyte in electronic devices. However, the performance of DSSC is still low using PEDOT:Carrageenan as electrolyte.

The first study of PEDOT:Carrageenan as electrolyte for DSSC obtained the highest efficiency of 0.42% using κ-carrageenan [33]. In this study, the synthesis process has been done by in situ polymerization with mixing EDOT monomer and κ-carrageenan solution. The different concentration of κ-carrageenan was used to investigate the effect of κ-carrageenan on PEDOT. The degraded-carrageenan was also prepared to observe the different molecular weight of carrageenan in PEDOT and its effects to conductivity value. Increasing the concentration of κ-carrageenan along with increasing the molecular weight enhanced the conductivity value and vice versa for degraded-carrageenan, decreasing the molecular weight of the carrageenan caused the conductivity value to be low. It also caused the performance of DSSC to be low.

In other studies, PEDOT:Carrageenan was synthesized using λ-carrageenan with a ratio of 1:1 [49] and κ-carrageenan with a ratio of 1:2 [50] in weight percent (wt%). Both showed low performance of DSSC. It was caused by the rigidity of PEDOT chain; electron transport was not as efficient as compared to the liquid system [49,50]. Both studies have synthesized the PEDOT:Carrageenan with a similar process via in-situ polymerization by mixing EDOT monomer and carrageenan solution. However, the conductivity of each sample was not measured, which was related to the DSSC performance.

## 5. Author’s Perspective

Polystyrene sulfonate (PSS) and carrageenan can be a dopant for PEDOT due to their ester sulphate group (-SO_3_) through a similar synthesis process that is oxidative polymerization. PSS and carrageenan are water-soluble polymers that assist PEDOT to be dispersed in many solvents. PEDOT:PSS as a counter electrode has been successfully combined with other materials in composite form to optimize the electrical conductivity and catalytic activity that can enhance the DSSC performance. PEDOT:Carrageenan as electrolyte on DSSC still obtained low performance. PEDOT:Carrageenan can probably be combined with other materials as well as PEDOT:PSS to enhance ionic conductivity and viscosity as electrolyte for DSSC to enhance the DSSC performance. PEDOT:Carrageenan as electrolyte polymer can be formed in gel polymer electrolyte, solid polymer electrolyte or composite polymer electrolyte.

PSS and carrageenan have differences. PSS has pH condition in acidic environmental, while carrageenan has pH condition in alkaline environmental. pH value could affect the degree of crystallinity of materials that could affect the electrical conductivity. Increasing pH value decreased the electrical conductivity due to the low degree of crystallinity. It can be one of the reasons that PEDOT:PSS can be used as counter electrode due to acidic pH that shows the high electrical conductivity. Meanwhile, PEDOT:Carrageenan has alkaline pH that shows the low electrical conductivity, which means PEDOT:Carrageenan can be used as electrolyte.

The main difference properties between counter electrode and electrolyte materials is electrical and ionic conductivity. As counter electrode, a material should have high electrical conductivity, while as electrolyte, a material should have high ionic conductivity. When a material has high electrical conductivity, it means that material has low ionic conductivity and vice versa. Since PEDOT:PSS has been successfully used as counter electrode for DSSC with high performance and the DSSC performance is still low with PEDOT:Carrageenan electrolyte, PEDOT:Carrageenan has a chance to be used as counter electrode also as well as PEDOT:PSS through some synthesis treatments, such as combined with other materials that can enhance the electrical conductivity and catalytic activity or decrease the pH value with added the acidic solution e.g., HCl, H_2_SO_4_, etc. during synthesis process.

## 6. Conclusions

The study of PEDOT:PSS as counter electrode and PEDOT:Carrageenan as electrolyte for DSSC application has been reviewed in this paper. PEDOT:PSS and PEDOT:carrageenan can be synthesized through similar synthesis process due to their similar ester sulphate (-SO_3_H) functional groups that assist PEDOT to be well dispersed in many solvents. The combination materials between PEDOT:PSS and metal oxide materials have enhanced the electrical conductivity and catalytic activity through increasing the roughness of film morphology and large surface area. This combination material has been successfully used as a counter electrode with highest DSSC performance of 8.49%. Meanwhile, the DSSC performance was still low by using PEDOT:Carrageenan as electrolyte. However, there is a chance to use PEDOT:Carrageenan as a counter electrode as well as PEDOT:PSS through some synthesis treatments with increasing the electrical conductivity than ionic conductivity of PEDOT:Carrageenan.

## Figures and Tables

**Figure 1 polymers-15-02725-f001:**
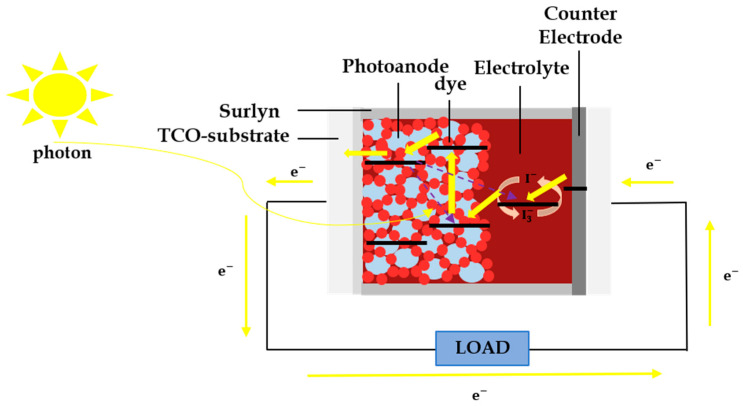
The working scheme of DSSC.

**Figure 2 polymers-15-02725-f002:**
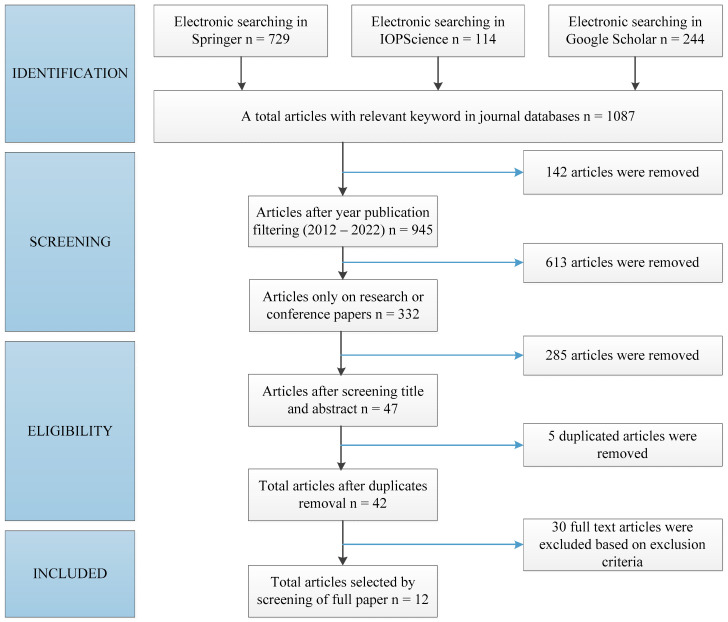
The PRISMA guidelines in selection procedures.

**Figure 3 polymers-15-02725-f003:**
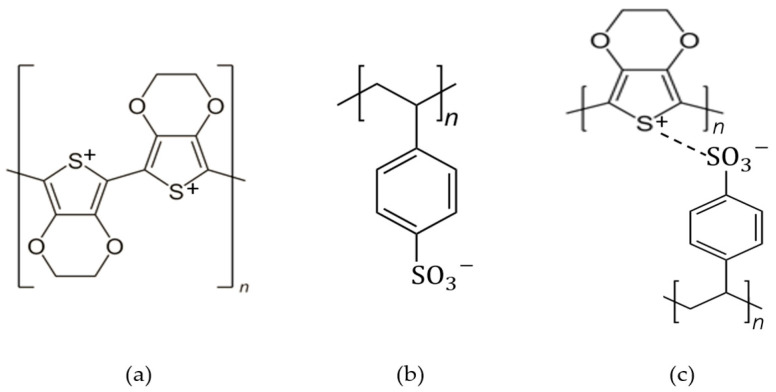
The chemical structure of (**a**) PEDOT, (**b**) PSS and (**c**) PEDOT:PSS.

**Figure 4 polymers-15-02725-f004:**
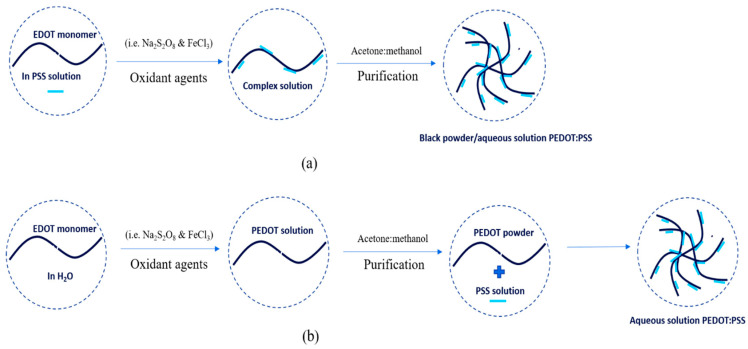
The synthesis process of PEDOT:PSS with (**a**) in-situ and (**b**) post-polymerization.

**Figure 5 polymers-15-02725-f005:**
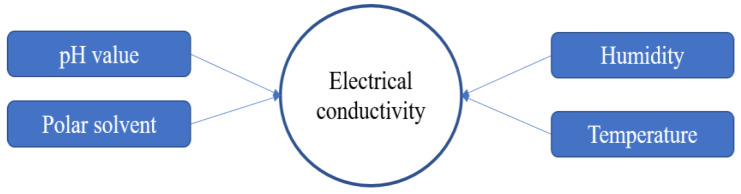
Factors affecting electrical conductivity.

**Figure 6 polymers-15-02725-f006:**
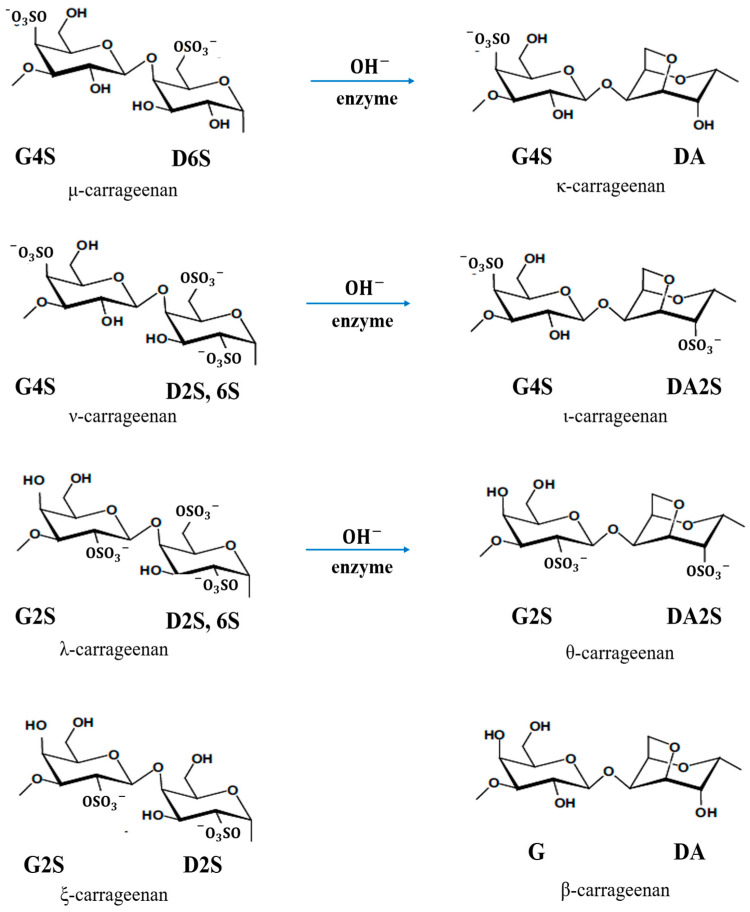
Chemical structure of all types of *carrageenan*.

**Figure 7 polymers-15-02725-f007:**
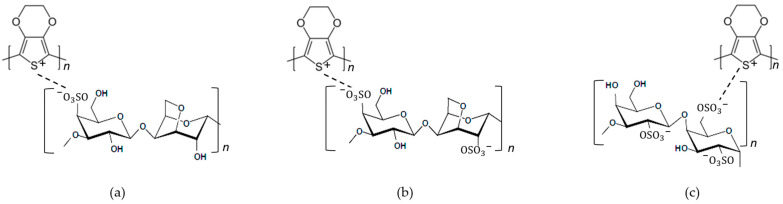
The chemical structure of (**a**) PEDOT:κ-Carrageenan, (**b**) PEDOT:ι-Carrageenan and (**c**) PEDOT:λ-Carrageenan.

**Figure 8 polymers-15-02725-f008:**
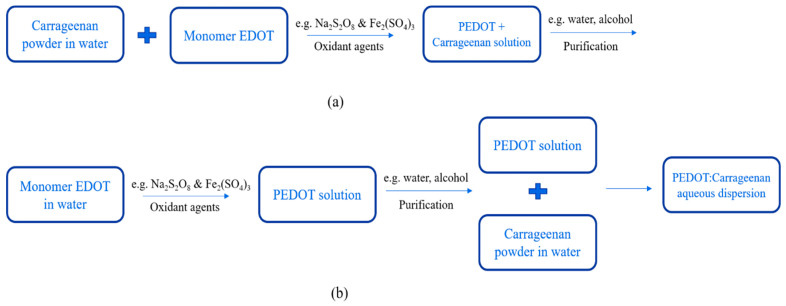
Synthesis process of PEDOT:Carrageenan with (**a**) in-situ and (**b**) post-polymerization.

**Table 1 polymers-15-02725-t001:** The identification database step.

Journal Database	Keyword	Results	Types of Documents
Springer	(“PEDOT:PSS” AND “counter electrode”) OR (“PEDOT:Carrageenan” OR “PEDOT:Carr” AND “electrolyte”) OR (“polymer conductive” OR “polymer electrolyte”) AND (“DSSC” OR “Dye Sensitized Solar Cell”).	729	Article, chapter, reference work entry, conference proceedings, conference paper, and reference work
IOPscience		114	Article and conference paper
Google Scholar		244	Article and conference paper

**Table 2 polymers-15-02725-t002:** FTIR analysis of PEDOT:PSS [18].

Wavenumber (cm^−1^)	Types of Atom Bonding	Identifications
691	C–S	Stretching of the thiophene ring in PEDOT
840		
936		
983	S–O	$\mathrm{Stretching}\;{SO}_{4}^{2-}$ from oxide and the S-phenyl bond in PSS
1145		Stretching in PSS
1055	S=O	$\mathrm{Stretching}\;\mathrm{antisymmetric}\;{SO}_{4}^{2-}$ from oxidant
1198		Stretching symmetric in PSS
1092	C–O	Stretching in PEDOT
1144		
1340	C–C	Stretching in the thiophene rings of PEDOT
1518	C=C	Stretching in the thiophene rings of PEDOT
1640		Stretching in the aromatic rings in PSS
2921	C–H	Stretching of PEDOT and PSS
3415	O–H	Stretching in PSS

**Table 3 polymers-15-02725-t003:** The letter code for the different sugar units found in *carrageenan* [26,27].

Letter Code	Carrageenan	IUPAC Name
D	Not Found	4-Linked α-D-galactopyranose
D2S	*ξ*	4-Linked α-D-galactopyranose 2-sulphate
D2S, 6S	*λ*,* υ*	4-Linked α-D-galactopyranose 2,6-disulphate
D6S	*μ*	4-Linked α-D-galactopyranose 6-sulphate
DA	*κ*,* β*	4-Linked 3,6-anhydro-α-D-galactopyranose
DA2S	*ι*,* θ*	4-Linked 3,6-anhydro-α-D-galactopyranose 2-sulphate
G	*β*	3-Linked β-D-galactopyranose
G2S	*λ*,* θ*	3-Linked β-D-galactopyranose 2-sulphate
G4S	*κ*,* ι*,* μ*,* υ*	3-Linked β-D-galactopyranose 4-sulphate
S	*κ*,* ι*,* λ*,* μ*,* υ*,* θ*,* ξ*	Sulphate ester (O-${SO}_{3}^{-}$)

**Table 4 polymers-15-02725-t004:** FTIR analysis of PEDOT:Carrageenan [14,26,30].

Wavenumber (cm^−1^)	Bond(s)/Group(s)	Letter Code	Type of Carrageenan
1495	C = C of thiophene ring	-	-
1371	C–C of thiophene ring	-	-
1198, 1060	C–O–C of stretching mode of the ethylene groups	-	-
892	C–H of PEDOT chains	-	-
1210–1260	S = O of sulphate ester	S	κ, ι, λ, μ, υ, θ, ξ
970–975	Galactose	G/D	κ, ι, μ, υ, θ, β
928–933, 1070 (shoulder)	C–O of 3,6-anhydro-D-galactose	DA	κ, ι, θ, β
890–900	Unsulphated β-D-galactose	G/D	β
840–850	C–O–SO_3_ of D-galactose-4-sulphate	G4S	κ, ι, μ, υ
825–830	C–O–SO_3_ of D-galactose-4-sulphate	G/D2S	λ, υ, θ, ξ
820, 825 (shoulder)	C–O–SO_3_ of D-galactose-2,6-sulphate	D2S,6S	λ, υ
810–820, 867 (shoulder)	C–O–SO_3_ of D-galactose-6-sulphate	G/D6S	μ
800–805, 905 (shoulder)	C–O–SO_3_ of 3,6-anhydro-D-galactose-2-sulphate	DA2S	ι, θ

**Table 5 polymers-15-02725-t005:** Electrical and optical properties of PEDOT:PSS and PEDOT:Carrageenan.

Properties	PEDOT:PSS	PEDOT:Carrageenan
Conductivity (S·cm^−1^)	>4000 [13]	16.23 [33]
Sheet resistance (Ω·sq^−1^)	<100 [13]	N/A
Transparency (%)	80–95 [13]	N/A
Ionic mobility (cm^2^v^−1^s^−1^)	2.2 × 10^−3^ [31]	N/A
Electronic mobility (cm^2^v^−1^s^−1^)	1.3 [31]	N/A
Carrier density (cm^−3^)	4 × 10^20^ [32]	N/A

**Table 6 polymers-15-02725-t006:** Performance of DSSCs utilizing PEDOT:PSS as counter electrodes.

Materials	Photoanodes	Dyes	Electrolytes	*A* (cm^2^)	*V_oc_* (volt)	*J_sc_* (mA·cm^−2^)	*FF*	*η* (%)	Ref.
PEDOT:PSS	TiO_2_/ZrO_2_	Z 907 (20 mg/L)	$\mathrm{Liquid}\;\mathrm{electrolyte}\;(I_{3}^{-}/I^{-}$)	0.5 × 0.5	0.61	5.04	0.29	1.77	[42]
PEG-PEDOT:PSS	TiO_2_	N3	$\mathrm{Liquid}\;\mathrm{electrolyte}\;(I_{3}^{-}/I^{-}$)	1 × 0.2	0.73	10.11	0.60	4.39	[43]
PEDOT:PSS-TiO_2_	TiO_2_	N719 (5 × 10^−4^ M)	$\mathrm{Liquid}\;\mathrm{electrolyte}\;(I_{3}^{-}/I^{-}$)	0.5 × 0.5	0.73	17.30	0.67	8.49	[38]
Ni-PEDOT:PSS	TiO_2_	N719 (5 × 10^−4^ M)	Organic electrolyte (T_2_/T^−^)	0.5 × 0.5	0.66	8.86	0.38	2.25	[44]
NiSO_4_-PEDOT:PSS	TiO_2_	N719 (5 × 10^−4^ M)	Organic electrolyte (T_2_/T^−^)	0.5 × 0.5	0.65	8.79	0.54	3.05	[44]
NiS-PEDOT:PSS	TiO_2_	N719 (5 × 10^−4^ M)	$\mathrm{Liquid}\;\mathrm{electrolyte}\;(I_{3}^{-}/I^{-}$)	0.5 × 0.5	0.76	16.05	0.67	8.18	[45]
PEDOT:PSS-Carbon	TiO_2_	N719 (0.5 mM)	$\mathrm{Liquid}\;\mathrm{electrolyte}\;(I_{3}^{-}/I^{-}$)	0.5 × 0.5	0.81	13.6	0.69	7.60	[3]
PEDOT:PSS-Graphene	TiO_2_	N719	$\mathrm{Liquid}\;\mathrm{electrolyte}\;(I_{3}^{-}/I^{-}$)	N/A	0.73	11.77	0.55	4.66	[46]
PEDOT:PSS/ SWCNH	TiO_2_	N719 (0.3 mM)	$\mathrm{Liquid}\;\mathrm{electrolyte}\;(I_{3}^{-}/I^{-}$)	0.5 × 0.5	0.68	14.06	0.52	5.10	[47]
MoS_2_/SWCNT-PEDOT:PSS	TiO_2_	Z 907	$\mathrm{Liquid}\;\mathrm{electrolyte}\;(I_{3}^{-}/I^{-}$)	0.4 × 0.7	0.78	16.21	0.64	8.14	[48]

**Table 7 polymers-15-02725-t007:** Performance of DSSCs utilizing PEDOT:Carrageenan as electrolyte.

Materials	Photoanodes	Dyes	Counter electrodes	*A* (cm^2^)	*V_oc_* (volt)	*J_sc_* (mA·cm^−2^)	*FF*	*η* (%)	Ref.
PEDOT:κ-Carrageenan	TiO_2_	N719 (3 × 10^−4^ M)	Pt	N/A	0.57	3.37	0.22	0.42	[33]
PEDOT:λ-Carrageenan	TiO_2_	N719	C/Pt	2.5 × 2.8	90.2 × 10^−3^	2.8 × 10^−3^	N/A	0.25 × 10^−3^	[49]
PEDOT:κ-Carrageenan	TiO_2_	N719	C/Pt	2.2 × 2.2	91.8 × 10^−3^	57.6 × 10^−3^	N/A	0.33 × 10^−3^	[50]

## Data Availability

Not applicable.

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
