# Peer review of "Different Roles between PEDOT:PSS as Counter Electrode and PEDOT:Carrageenan as Electrolyte in Dye-Sensitized Solar Cell Applications: A Systematic Literature Review"

_polymers, 2023, doi:10.3390/polym15122725_

Round 1
Reviewer 1 Report
In this manuscript, the author reviewed PEDOT:PSS and PEDOT:Carrageenan, which includes their synthesis and properties. First, the English should be carefully polished before publication. Several suggestions are listed below:
1. At line 28, the name “Gratzel” should be revised to “Grätzel” or “Graetzel”.
2. At line 72, “Wei Wei” should be revised to “W. Wei”.
3. At line 74, “etc” should be revised to “etc.”
4. It is better to include a table of electrical and optical properties of PEDOT:PSS and PEDOT:Carrageenan. The properties could be electrical conductivity, mobility, carrier density, ionic conductivity, transparency, etc.
5. At line 155, does “PEDOT:PSS was first synthesized by scientist research laboratories in German” mean “PEDOT:PSS was first synthesized by scientists in German research laboratories”?
6. At page 7, the discussion should mention exact data of PEDOT:PSS. For example, how much did the conductivity decrease when adding a certain amount of NaOH?
7. At line 338, “Jsc” should be italic. Other variables should be italic, in the rest part of this manuscript.
8. Abbreviations should be cited at its first appearance. E.g. TiO2 at line 353.
9. At line 296, “PEDOT:PSS is one of the counter electrode polymers that has been used in DSSC with a high performance of ~10% [30].” This work should be cited in Table 5.
The English should be carefully polished.
Author Response
Dear Reviewer,
Thank you for the suggestions and comments. We have provided a point-by-point response to your suggestions and comments in a file. Please, see the attachment.
Thank you.
Best regards,
Authors

Reviewer 2 Report
This paper is well written and of interest.
This article is well written and interesting.
However, I think the authors should add a final table summarising the different properties of the two types of PEDOT (including: typical mobility values, pH, molar weight of polymers, solubility, absorbance, ...).
Author Response

(The authors gave the same response as above.)

Reviewer 3 Report
In this review manuscript, titled "Different Roles of PEDOT:PSS as a Counter Electrode and PEDOT:Carrageenan as an Electrolyte in Dye-Sensitized Solar Cell Applications: A Systematic Literature Review," the authors specifically discuss the synthesis, characterization, and application of PEDOT:PSS and PEDOT:Carrageenan in DSSCs. While the manuscript is well-written and easy to follow, it lacks sufficient information and detail to be published in Polymers in its current version. Therefore, I would recommend a major revision. Below, you will find my comments for improvement:
1. The format of a review paper differs from that of a research paper. The "Materials and Methods" section is typically used in research papers to discuss the sample preparation and characterization procedures. I would recommend that the authors closely follow Polymers' review format. The paper selection session can be moved to the supporting information section, as it does not contain scientific content and may not be of interest to readers.
2. On page 5, Table 2, I recommend adding the chemical structure of PEDOT:PSS and highlighting the bonding within the structure. An example can be easily found online, such as in Figure 5 of the article https://pubs.acs.org/doi/pdf/10.1021/acsapm.9b00757.
3. On page 9, Table 4, I suggest making the same recommendation as for Table 2 on page 5.
4. In Section 3.2, "Synthesis and Characteristics of PEDOT:Carrageenan," it is critical to compare the conductivity, measured in S/cm, across different types of samples.
Easy to follow but contains typos.
Author Response

(The authors gave the same response as above.)

Round 2
Reviewer 1 Report
The author addressed all comments appropriately. I recommend to accept it.
Reviewer 3 Report
The authors have addressed my concerns, this manuscript is recommended to be accepted in the current version.
The content is easy to follow.